# Spatial Video and EpiExplorer: A Field Strategy to Contextualize Enteric Disease Risk in Slum Environments

**DOI:** 10.3390/ijerph19158902

**Published:** 2022-07-22

**Authors:** Jayakrishnan Ajayakumar, Andrew J. Curtis, Vanessa Rouzier, Jean William Pape, Sandra Bempah, Meer Taifur Alam, Md. Mahbubul Alam, Mohammed H. Rashid, Afsar Ali, John Glenn Morris

**Affiliations:** 1Department of Population and Quantitative Health Sciences, School of Medicine, Case Western Reserve University, Cleveland, OH 44106, USA; ajc321@case.edu; 2Les Centres Haitian Group for the Study of Kaposi’s Sarcoma and Opportunistic Infections (GHESKIO), Port-au-Prince HT6110, Haiti; vrouzier@gheskio.org (V.R.); jwpape@gheskio.org (J.W.P.); 3Department of Geography, Kent State University, Kent, OH 44240, USA; sbempah@kent.edu; 4Emerging Pathogens Institute, University of Florida, Gainesville, FL 32601, USA; mtalam@epi.ufl.edu (M.T.A.); md.alam@epi.ufl.edu (M.M.A.); mhrashid@epi.ufl.edu (M.H.R.); afsarali@epi.ufl.edu (A.A.); jgmorris@epi.ufl.edu (J.G.M.J.); 5Department of Medicine, College of Medicine, University of Florida, Gainesville, FL 32601, USA; 6Department of Environmental & Global Health, College of Public Health and Health Professions, University of Florida, Gainesville, FL 32601, USA

**Keywords:** spatial video, exploratory analysis, geospatial context, mobile methodologies

## Abstract

Disease risk associated with contaminated water, poor sanitation, and hygiene in informal settlement environments is conceptually well understood. From an analytical perspective, collecting data at a suitably fine scale spatial and temporal granularity is challenging. Novel mobile methodologies, such as spatial video (SV), can complement more traditional epidemiological field work to address this gap. However, this work then poses additional challenges in terms of analytical visualizations that can be used to both understand sub-neighborhood patterns of risk, and even provide an early warning system. In this paper, we use bespoke spatial programming to create a framework for flexible, fine-scale exploratory investigations of simultaneously-collected water quality and environmental surveys in three different informal settlements of Port-au-Prince, Haiti. We dynamically mine these spatio-temporal epidemiological and environmental data to provide insights not easily achievable using more traditional spatial software, such as Geographic Information System (GIS). The results include sub-neighborhood maps of localized risk that vary monthly. Most interestingly, some of these epidemiological variations might have previously been erroneously explained because of proximate environmental factors and/or meteorological conditions.

## 1. Introduction

Household access to safe drinking water, uncontaminated food, appropriate sanitation, and the removal of vector causing trash are arguably the biggest challenges faced in slum/informal settlement (SIS) environments [1,2]. The spatially granular identification of these risks on a sustainable continuous basis has previously been identified as a vital aspect of epidemiological/public health monitoring. For example, the contamination of water points with fecal matter [3,4] can occur because of various (often overlapping) dynamic risks based on the site (the location of the water point) and its situation (the environmental and behavioral factors present in its immediate vicinity). Even improved water sources, such as underground pipes, can become contaminated through natural disasters and human-driven illegal tapping [5], whereas the containers sunk into the ground to hold these reservoirs for neighborhood consumption can become contaminated by spillover flow from nearby drainage channels [6,7]. Temporal risk, therefore, varies because of both human agency and environmental/metrological fluctuations. Heavy rains can direct the flow from the drainage ditch into the reservoir [8], contaminated water could leach through porous walls, or people could simply rest their containers in the surrounding mud, which may also contain fecal matter, before being lowered into the water point.

Being able to collect the right data that can reveal spatial and temporal variations in this landscape of risk is vital, not only to understand where and why risk changes, but also to be able to provide early warning systems if a particular water access point becomes unsafe [9,10]. Previous work by the authors in the area of mobile data collection using spatial video (SV) and its geonarrative variants have advanced the science of gaining a contextualized understanding of localized risk [11,12,13,14,15,16,17]. However, though some of this work has previously been used to characterize the nature of risk around and in water points in Haiti [13], an identified deficiency has been the flexibility to reveal space–time patterns in these locations. The typical mapping of locally collected risk factors still relies on more traditional spatial software, such as a Geographical Information System (GIS). For example, if “X” water points generated fecal coliform samples and associated environmental variables, then more traditional GIS analysis will either be cross-sectional snapshots (in which water points are elevated for this period) or a search for generalized space–time associations (such as an increase in rainfall leading to fecal coliform increases). This type of traditional GIS uses fails to reveal how a combination of variables influence each other at a single site, and then between sites, and across multiple time periods.

In this paper, we address this localized mapping conundrum by drawing on the types of data analytics required to support a more spatially exploratory approach. During the authors’ spatial response to COVID-19, the power of real-time data analytics and “big data” investigations, especially the use of dashboard style data manipulations, led to a fresh perspective on epidemiological “space-time risk scenarios” [18]. Though other interactive software is available to interactively explore data [19], including different dashboard applications, which allow for the inclusion of spatial data [20,21,22], none have the location-specific sophistication and flexibility needed to fully leverage the different data types generated here. In this paper, we show how exploratory spatial data analytics utilizing bespoke spatial programming can be used to more effectively answer (epidemiological) questions based on the types of data available, interactive multi-media visualizations, and the environment being studied. To illustrate the power of this approach, we explore the complex interactions in the epidemiological, environmental, and human behavior risks around water access and drainage locations for three coastal informal settlements in Port-au-Prince, Haiti.

## 2. Design and Functionality

Spatial programming and database construction are becoming more commonplace in geographic investigations. These information science skills, along with other approaches to mine large data sets, such as using machine learning, have opened different ways to consider data [23]. In Haiti, the team has been collecting epidemiological and environmental data for over five years. The resulting information consists of various data types, including textual data, video images, and spatial data (in the form of GPS paths). New ways to automate the digitizing of environmental risks from SV images using machine learning are currently in development [24].

To fully leverage these different data types to understand the question of how water risk varies in space and time as part of a complex dynamic landscape of interaction features, we had to develop a tool that could replace the existing, more limited (spatial) software. Simply put, the tasks required of the approach we developed here (and what we will call EpiExplorer for simplicity’s sake) evolved out of the types of field epidemiological questions that needed to be asked, e.g., which water points became a health risk and why? To achieve this goal, we had to combine the epidemiological and environmental inputs utilizing a database that supports various exploratory analytic and visualization techniques (Figure 1).

### 2.1. Input Module

In order to fully understand how the health risk associated with localized water access, a map of all water points was created (entered as a coordinate or by simply clicking on the map and adding a marker). For each location, a series of attributes were attached, such as category type (whether it was a pipe or cistern). In addition, other environmental testing locations, such as ocean sites or drainage channels, were added in the same way. Other media, including still and video images, were also attached to these locations to provide temporally varying context. More complex epidemiological (microbiological and physiochemical water testing) data were then added, such as temperature, dissolved oxygen content (DO), salinity, pH (indicating the level of acidity), fecal coliform count, and the date of collection.

As previous studies have suggested that the human–environment activity around each water point can also add localized risk, an environmental survey was also added to the database. The environmental survey includes the level of mud around the water point, the amount of proximate standing water, trash (plastic and biological waste), and activity (movement of people around water point), all digitized from the spatial video collected during each water sample collection. These environmental data were categorized on a scale of 1 to 5, with 1 indicating the least amount. This meant that the proximate environmental and observed activity could be matched to the epidemiological data.

### 2.2. Database Module

In order to facilitate querying, a local database was built to interact with both the input and exploratory module to synchronize the process of seamless data integration and visualization. The table “water_point” contains details of all the water points. Apart from storing details such as the neighborhood where the water point is located, the category (pipe/cistern, ocean, and drainage), the address of the location, and the image for the water point, the water point table also stores the coordinates of each water point. Whereas the water points can be queried based on textual attributes such as category, the coordinates can be used to perform spatial queries. The epidemiological and environmental data are stored in separate tables (Figure 2A,B), with each containing numerical (temperature, salinity, pH, mud, trash, activity), date (when the records where collected), and media (spatial video images) attributes. These attributes can be used to perform various textual, numerical, and temporal queries.

### 2.3. Exploration Module

All sample water points added to the database are spatially visualized as marker pins in Google Maps (Figure 3A). This is helpful for users with minimal or no previous experience with a GIS. Hovering on top of the water point will display its basic details, as well as its image. This type of simple visualization was found to be important, as many of our in-field collaborators could then easily explore the mapped area, and participate in collaborative discussions irrespective of where they are located. The parameters required for space–time querying can be retrieved by clicking on the marker (Figure 3B); selecting, for example, the date of interest; and then using “*Visualize*” to explore different temporal patterns for that water point. The environmental variable time-chart (Figure 4) shows the longitudinal variations in various environmental factors for the selected water point. Hovering on top of each data point also displays the associated environment data, illustrated here for mud (Figure 4B), water (Figure 4A), trash (Figure 4C), and activity (Figure 4D). The red data point (Figure 4) indicates the current query (corresponding to the selected date), which can be switched simply by clicking on any other data point, making it the active search.

Another dynamic chart displays the percentage of coliform count for a water point for the selected period with respect to the maximum coliform count for the same water point (Figure 5B). As an example, if the maximum fecal coliform count for a particular water point is 100, but for the selected date, the fecal coliform count is 20, then the percentage fecal coliform count will be 20%. In this way, it is immediately obvious to see how problematic that location is based on its previous life course. Hovering on a data point also displays details such as fecal coliform count value, fecal coliform percentage, and the date for which the sample was collected. This allows for an easy interactive investigation of the outlier samples. The relative risk of a water point can also be assessed against those other tested locations in the same neighborhood (Figure 5A). As such, the water point’s risk can be viewed as both a snapshot of its own history and its overall current situation. Using the dynamic rank-chart, outlier water points can be identified (extremely high or low values of experimental variable compared to other water points) and further explored for localized issues. The rank-chart (Figure 5A) is interactive, and a left-click on any data point will generate another query for the newly selected water point for the same period. A right-click on any data point will animate the corresponding water point on the map, which helps to contextualize the data with its surrounding geography.

A vital advance was in the ability to spatially explore both epidemiological and environmental data. For example, Figure 6A displays the spatial distribution of fecal coliform counts across water points for the same neighborhood. The water points are rank-ordered based on the fecal coliform count, and displayed using proportional symbols. The water point under investigation is indicated with a red fill color, whereas all other water points are green in color. For the Haiti example, though three different named collection areas were part of the study, their proximity to one another also meant cross-area comparisons were useful. To perform this, any non-neighborhood water point could be added (by clicking on the location), which leads to a reordering and redrawing of the proportional symbol map. Clicking on any water point in the map also refocuses all investigations on to that one location. In this way, these data can be explored spatially to see, for example, how similar the attribute values are for two neighboring water points. The environmental variables can be spatially investigated in the same way (for example, mud, standing water, etc.).

Though the visual exploratory tools are useful for user-driven exploratory investigations, it is also important to include more traditional overview spatial summaries to help guide this process. To this end, we decided to include a dynamic heat map (Figure 7), along with a time slider, to visualize the spatial changes in a variable’s intensity across the map for any period. Any variable can be mapped in this way to show the stability of hotspots, or how they migrate across the study region. In this way, a pseudo-animation can be created by having the user move the slider bar back and forth. To illustrate this, Figure 7 shows the coliform count density for the M neighborhood for February 2018, revealing one water point with a high intensity requiring further investigation.

Though EpiExplorer supports multiple interactive spatio-temporal querying, production cartography and other forms of spatial analysis are still performed within a traditional GIS (such as ArcGIS or QGIS). To support continuous integration with these types of tools, data can be download in shapefile and comma-separated value (CSV) formats. In this way, the initial exploratory investigations can be conducted using this bespoke approach, though with outputs and findings easily being displayed within more commonly used software.

To illustrate the epidemiological advantage such a bespoke approach has, we will use a case study of monthly environmental and epidemiological records collected for thirty-eight locations across three different neighborhoods of Port-au-Prince in Haiti [13].

## 3. Case Study: Spatio-Temporal Variations in FC Count

### 3.1. Data

Beginning in October 2016, water samples were collected every month in three neighborhoods (S, M, and P) of Port-au-Prince [13]. Field researchers visited each testing location to draw water samples, with temperature, turbidity, total dissolved solids, pH, and dissolved oxygen (DO) being recorded in situ. The samples were further tested on returning to the laboratory to measure the fecal coliform count (FC) using the membrane fecal coliform (mFC) agar method. These data were then added to EpiExplorer as one of the following categories: “Pipe/Cistern” or “Drainage” or “Ocean Site” or “Combined”.

In addition to the epidemiological data, the field team also collected micro environmental surveys around each sample location using Contour +2 and Patrol Eye Body cameras [13]. These spatial video (SV) cameras have an in-built global positioning system (GPS) receiver which geotags each image frame. Researchers at the GIS Health and Hazards Lab utilized these videos to digitize environmental risks, including “mud”, “water”, “trash”, and “activity” [14]. Risks were assigned a score from 1 to 5, with 1 and 5 indicating low and high values, respectively, for all months where SV were collected. An example image for each water point *for each collection period* was also added to assess how any testing site might have physically changed across time.

### 3.2. Longitudinal Variation in FC Count

Once these data were uploaded, various exploratory investigations could occur. To illustrate this, three different examples are presented where the micro-environmental epidemiological understanding and the localized risk profile were enhanced due to the investigation. The epidemiological data time-chart for the M13 water point (Figure 8A) shows a single large spike for July 2017, with an FC count of 35,800,000 as compared to 0 in all other months. To investigate this anomaly, the environmental data time-chart was queried (Figure 8B). For the month of July, the environmental variable “water” was also high around the water point. By comparing water conditions around the water point using the SV imagery, we verified that this month was indeed the wettest (and muddiest) (Figure 9D) when compared to the other months, such as January (Figure 9A), February (Figure 9B), March (Figure 9C), October (Figure 9E), and December (Figure 9F). Indeed, for these other periods, the ground appeared relatively dry. (One obvious dataset not included here is rainfall [25]; drainage systems could overflow and contaminate the area around a water point, or even enter directly into the water supply through fecal matter transport. However, little to no actual meteorological data are available for the study regions described in this paper. As an alternative, we did use rainfall data provided by the European Center for Medium-Range Weather Forecasts (ECMWF) [26] (Figure A1A). The details of the experiment are provided in Appendix A. The results suggest that there is no relation (correlation coefficient of −0.098) between the variables of FC count and rainfall (Figure A1B). If we remove the FC count for M13 for the month of July (Table A1), which seems to be an outlier, there seems to be a weak positive correlation (0.46) between FC and rainfall. However, we should be careful about interpreting the results from this experiment, as the rainfall data are available only at a coarser level of granularity. However, even after rain events, the micro-geography of a water point will still cause variation in how wet the ground remains).

### 3.3. Micro-Scale Spatial Variation in FC Count

The FC count for M13 was particularly high for July 2017 (Figure 8A. To provide geographic context for this finding, we utilized the heatmap (Figure 10), which revealed that this count was not only anomalous within the water point’s own temporal history, but also with respect to the other test locations in that period. The proportional symbol map for the FC count (Figure 11A) tells the same spatial story, whereas the proportion symbol map for the environmental variable, “water” (Figure 11B), also reveals a higher value than for the surrounding neighbors.

To explore what was happening at this water point, we consider M1, its immediate neighbor approximately 30 m away. The epidemiological time-chart for M1 (Figure 12A) has a FC count of zero for the same period. The SV imagery for M1 during July (Figure 13A) also shows the conditions around the water point as being dry when compared to M13 (Figure 13B) for the same period.

Sometimes variations in the results are not due to environmental factors. For example, a spike in FC counts at S17 occurred in May (FC count = 3,868,250), June (FC count = 890,000), and July (FC count = 209,800) in 2017 (Figure 14A). Compared to these three months, the next highest FC count is 855 for November 2017. Obviously, something important had occurred at this water point during these three months. The epidemiological map for the S neighborhood for May (Figure 14B), June (Figure 14C), and July (Figure 14D) shows that S17 was also somewhat of a spatial anomaly during this period when compared to its immediate neighbors. By switching to the corresponding images for May (Figure 15D), June (Figure 15E), and July (Figure 15F), water samples were collected from a pipe (Figure 15F), but when we compared the images for February (Figure 15A), August (Figure 15B), and September (Figure 15F), we found that, for these months, the water samples were collected directly from the adjacent tank that was placed inside an elevated concrete housing. The associated FC count was 100, 98, and 855, which is relatively low when compared to the three-month spike.

Similar to the previous example, at P1, there was a steep increase in FC count for May (FC count = 33,000) and June (FC count = 1,050,000) 2017 (Figure 16). These values are in marked contrast to October (FC count = 0), November (FC count = 2), and December (FC count = 0). Again, by referring to the SV images for these months, we see that for May (Figure 17A) and June (Figure 17B), the water samples were collected by dipping the water sample container directly into the water tank. By contrast, in October (Figure 17C), November (Figure 17D), and December (Figure 17E), the water samples were collected from the tap that was connected to the tank.

## 4. Discussion

Safe water access in many developing world environments is not simply a distance-based measure of accessibility, but rather a localized function of water point type and the interconnection between the water source and other features, such as drainage [13,27,28]. Adding further complexity is human activity and its interaction with the environment. Understanding how water-related risk varies at a sub-neighborhood scale so that longer-term changes and improvements can be made requires multiple types of data, all of which will interact with each other in space and time. Though traditional mapping can be used to understand patterns in space and time, it is often limited by the data available and the flexibility of traditional spatial software to tease out patterns at a meaningful scale to capture local processes.

In this paper, we have used a data analytics approach to more fully leverage fine-scale longitudinal epidemiological and environmental risks. Using new field-based methods, such as spatially-encoded video, local visual data have been collected to both capture the quality and type of the testing location, and also the proximate environmental risks. If we assume that each testing location is unique—a product of its own setting—then the number of data combinations involved in any investigation quickly grows; how does one water point change across time, in comparison to itself or to other sites collected at the same time? Are there spatial patterns in those changes?

Being able to move easily between locations and between data inputs allows us to fully explore these types of questions. More importantly, these changes can be investigated dynamically using data-driven and cartographic tools. Though more traditional hotspot analysis might reveal a general pattern for one time period, or even a developing trend across multiple time periods, these methods miss questions such as: why does a hotspot/cold spot pattern fluctuate between two time periods, and is that same pattern echoed elsewhere, and with other variables? In other words, there is more space and time nuance in real-world settings.

For example, from the temporal analysis of the FC count for M13, we found that the immediate surrounding of M13 was visibly waterlogged (Figure 9D) for the month of July, which could account for FC growth [29,30,31]. The suggestion here is that the proximate effect of the water led to the contamination of the water supply, a risk often described in similar setting studies, but rarely validated [32]. The comparison of two nearby water points, M13 and M1 (Figure 12), revealed the heterogeneity in environmental conditions across short distances. It is likely these water points experienced similar rainfall, as they are separated by only 30 m, but their environmental risks vary because of soil type, construction, and proximity to drainage. This shows how environmental conditions can vary dramatically even across relatively short distances, which also means the relative risk varies spatially. This is a problem if data are only available for neighborhood scales or even coarser scales, e.g., extrapolated meteorological data. Though adding in finer-scaled meteorological data would be beneficial, it is likely that these data would not be available at the granularity described here. Even so, if we knew that the entire area of Port au Prince had suffered high rainfall, exploring the previous months patterns of environmental risk using EpiExplorer might help suggest where problems might emerge.

We have shown that micro spaces “matter” using these various dynamic cartographic and analytical tools, identifying various spatial anomalies, and then searching for causations. In another example, the effect of environmental conditions on FC count can also be gleaned from the analysis for water point S17 (Figure 14). For the collection periods of May, June, and July (Figure 15), access changed to the pipe that was outside the concrete housing. We do not know whether the pipe itself was contaminated or whether spillover effects from the muddy ground were to blame, but without the utility of EpiExplorer (and access to the spatial video imagery), other environmental factors might have been proposed to explain this anomaly, whereas the answer appears to be how and where the water sample was taken for the same general site. The resulting fluctuation in the FC count might have prompted further investigation into explanatory environmental and meteorological causations, but these, in this case, would have missed the real underlying reason. Similarly, in another example, for water point P1, the explorer helped explain FC anomalies due to a variation in where the water was sampled (Figure 17). The images suggest that the water collection location was not consistent and could be a potential source of variability in the FC counts for different periods. The follow-up question is: why is this second water access point so much more problematic?

This is not a single period or single site analytical tool. Now that a baseline has been established, other sites can be added, and other time periods can be compared. Indeed, data collection for other neighborhoods in Haiti continued through 2021, with these new neighborhoods and testing locations being added and ready for future analysis. If, for example, there was a major enteric disease outbreak in any of these neighborhoods, then we could return to gauge previous micro-space risks in the impacted area, and potentially help guide an intervention in what would be the most vulnerable areas.

Though a major goal of EpiExplorer was to more fully leverage the types of data being collected by the Haiti field teams, it was also designed for other uses on Windows-based stand-alone computers. The code, along with the standalone software, can be downloaded from https://github.com/JayakrishnanAjayakumar/EpiExplorerSoftware, accessed on 30 May 2022. We also plan to create a web version of the software, so that the software is not tied to any particular operating system. A limitation with this type of bespoke spatial software is the issue of transferability. It is likely that the types of questions asked in this paper, and the exploratory displays presented, will also be relevant for similar environments. Therefore, a key concept in EpiExplorer’s development was replication in other environments, where other researchers could upload similar multi-site, multi-time-period, and multiple variable input. Currently in the Democratic Republic of Congo, a similar tool is being used to investigate patterns of risk around Goma.

Future versions could benefit from expanding the database and providing the user with the opportunity to create database tables seamlessly. The map and chart-based visualizations can also be extended to support spatio-temporal visualization, e.g., using more traditional methods, such as a localized version of Martin Kulldorff’s spatial scan statistics [33]. Apart from images, the support for adding other media sources, such as short video clips, can also be beneficial for improving contextual analysis. Finally, as with ongoing research in other environments, adding textual or narrative data would provide even more insights [34,35].

## 5. Conclusions

Linking health risks to spatially granular causative processes is a recognized need in socially vulnerable settings, such as slums or informal settlements. These risk settings are also likely to vary across time periods due to seasonal, meteorological, environmental, or behavioral changes. Though new geospatial technologies now provide the means to perform on-the-ground surveys to collect that data, the means to analyze and visualize patterns to fully leverage insights at a hyper local level are still lacking. To address this gap, we utilized data analytics approaches, including spatial programming, to develop a new set of interactive tools that allow researchers to investigate such nuances within a space–time risk landscape. The results revealed how risk anomalies could be identified. Some of these could explained by typical environmental patterns, such as the presence of local standing water, but in other situations, the causations were more complicated and behavioral in nature. These required further mining of the available data. What we have shown though, is that this combination of new geospatial field tools and the means to display and analyze the resulting data allow for such a nuanced investigation. This means that epidemiological research in challenging environments can start to move from conceptualizations of risk to actual local research questions that can explored and tested, while new theoretical frames can be developed.

## Figures and Tables

**Figure 1 ijerph-19-08902-f001:**
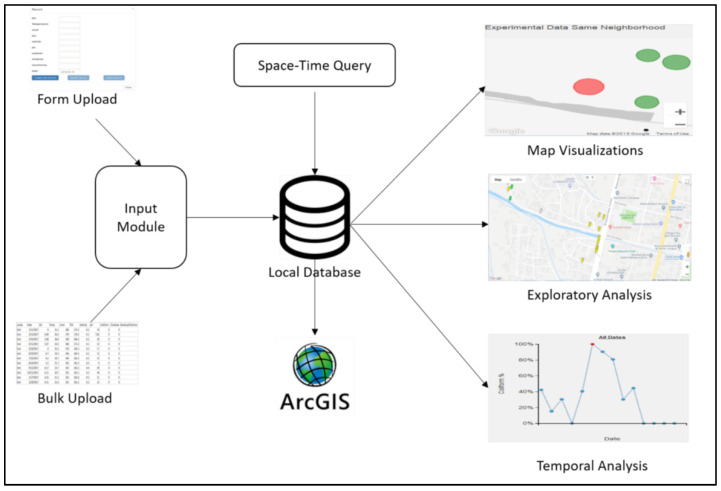
Schematic diagram for EpiExplorer with input, database, and exploratory module.

**Figure 2 ijerph-19-08902-f002:**
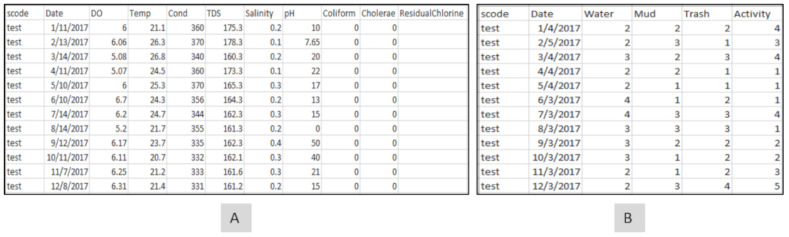
Sample database records for (**A**) experimental and (**B**) environmental records.

**Figure 3 ijerph-19-08902-f003:**
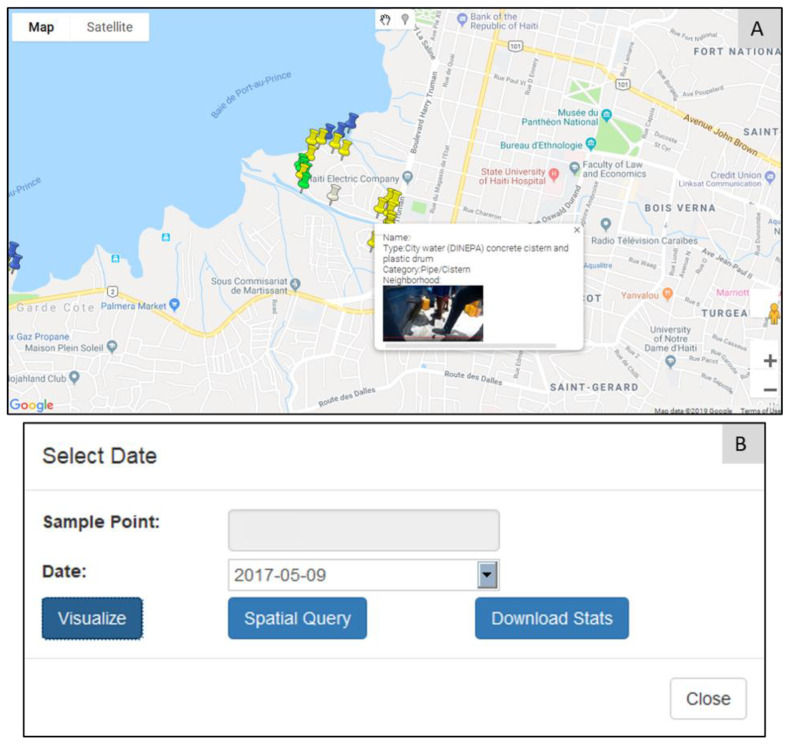
Water point visualization and parameter selection. (**A**) Water points represented as marker pins; (**B**) Clicking on a marker displays the visual parameters. The yellow, green, blue, and white markers indicate pipe, drainage, ocean, and combined water point types, respectively.

**Figure 4 ijerph-19-08902-f004:**
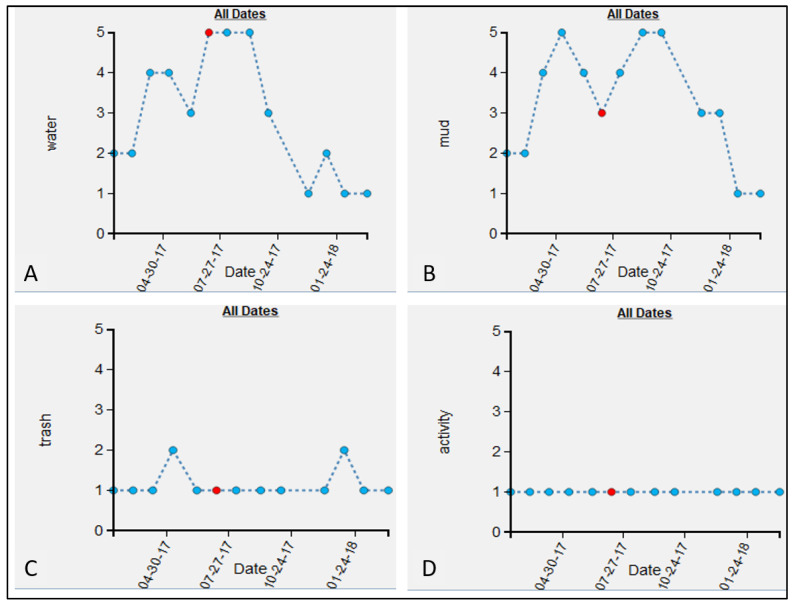
Environmental variable time-chart for (**A**) water, (**B**) mud, (**C**) trash, (**D**) activity. The red dot indicates the currently queried instance.

**Figure 5 ijerph-19-08902-f005:**
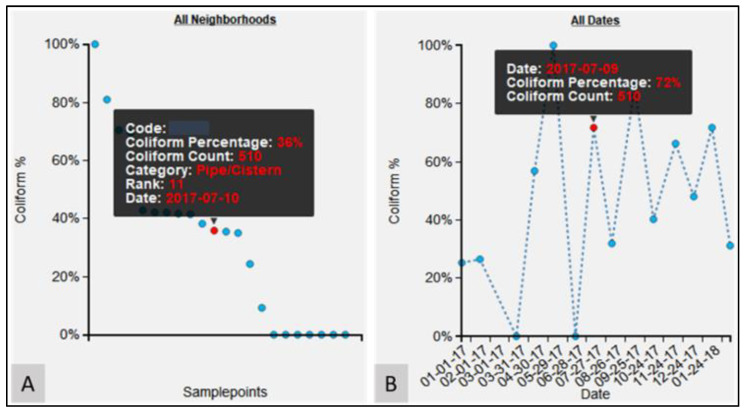
Visualization charts for epidemiological variables. (**A**) Rank-chart for all neighborhoods; (**B**) Time-chart showing longitudinal variation for the particular epidemiological data.

**Figure 6 ijerph-19-08902-f006:**
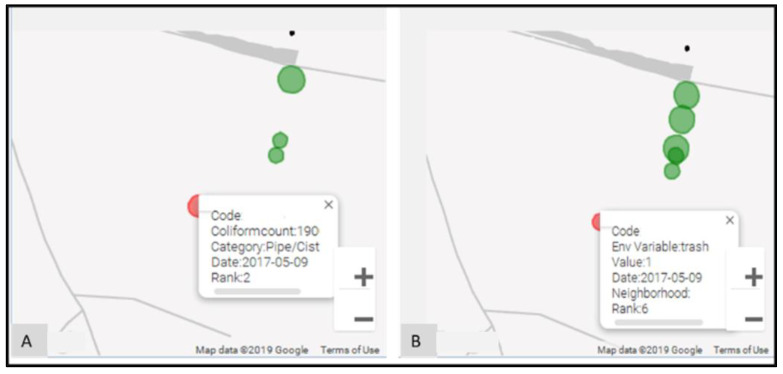
Proportional symbol map to display the spatial distribution of: (**A**) epidemiological records, (**B**) environmental records.

**Figure 7 ijerph-19-08902-f007:**
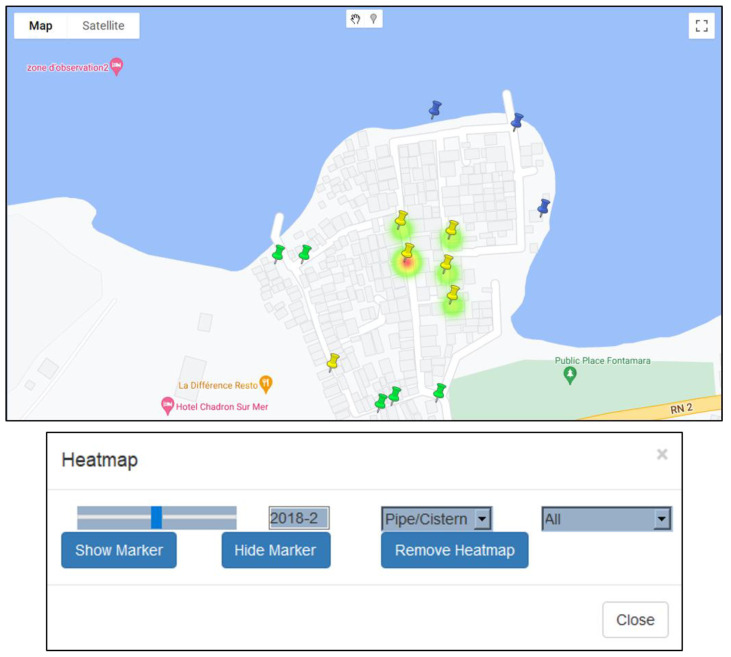
Heatmap displaying the overall spatial distribution of epidemiological records.

**Figure 8 ijerph-19-08902-f008:**
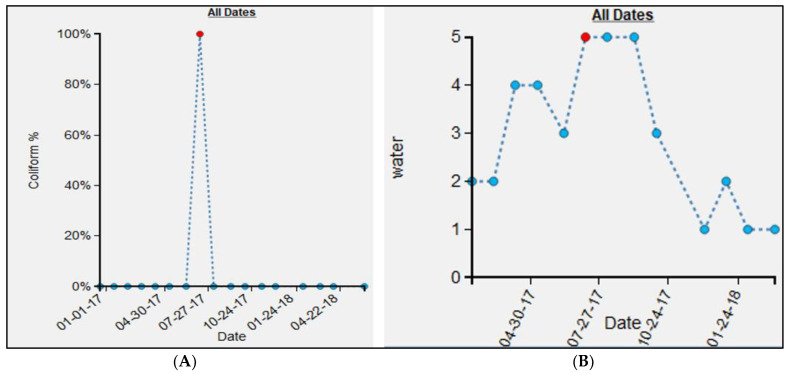
Epidemiological and environmental record for M13 water point: (**A**) time-chart for FC count, (**B**) time-chart for the environmental variable water.

**Figure 9 ijerph-19-08902-f009:**
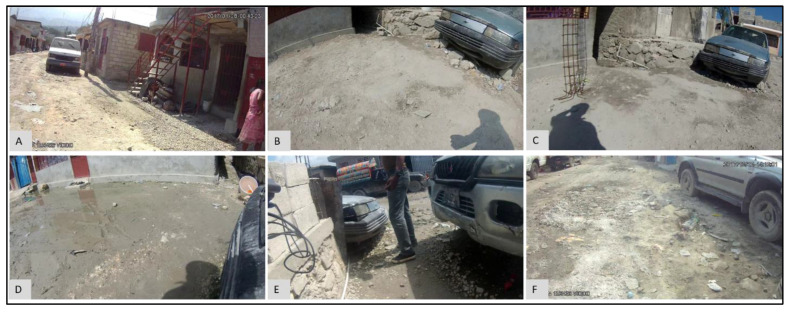
SV image frames for M13 water point for: (**A**) January, (**B**) February, (**C**) March, (**D**) July, (**E**) October, (**F**) December.

**Figure 10 ijerph-19-08902-f010:**
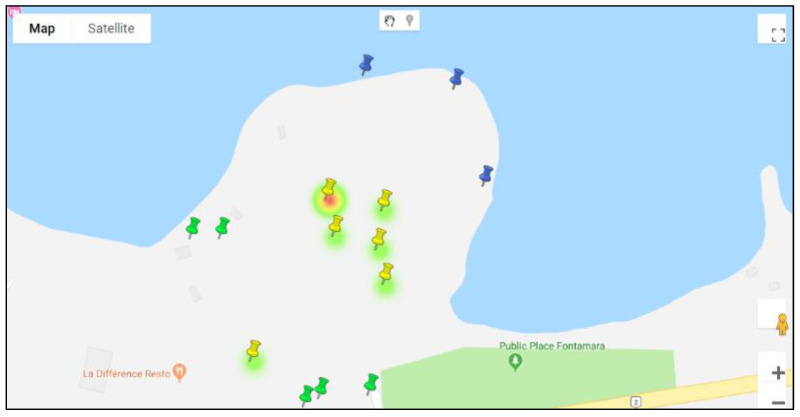
Heatmap for M neighborhood for July 2017. The water point with the highest intensity is M13. The water point south of M13 (nearest) is M1.

**Figure 11 ijerph-19-08902-f011:**
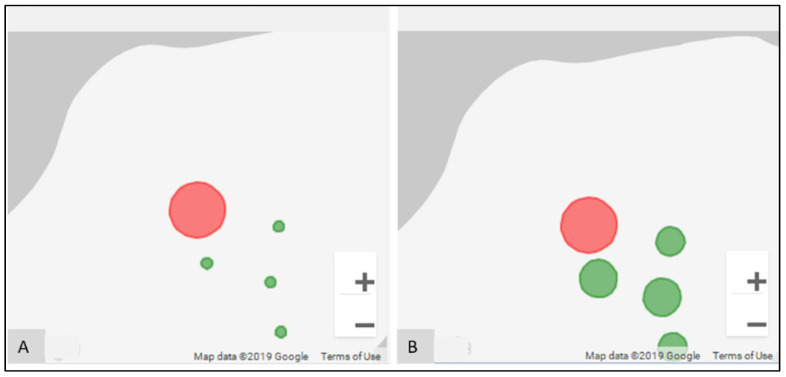
Proportional symbol map for July 2017 for M neighborhood for: (**A**) FC count; (**B**) environment variable, “water”.

**Figure 12 ijerph-19-08902-f012:**
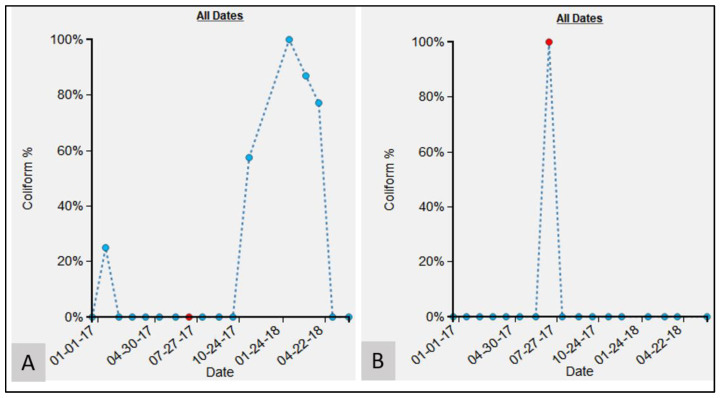
Time-charts for epidemiological variable FC count for: (**A**) M1, (**B**) M13.

**Figure 13 ijerph-19-08902-f013:**
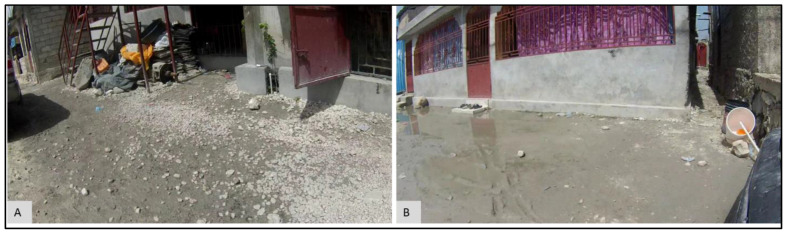
SV imagery for July 2017 for: (**A**) water point M1, (**B**) water point M13.

**Figure 14 ijerph-19-08902-f014:**
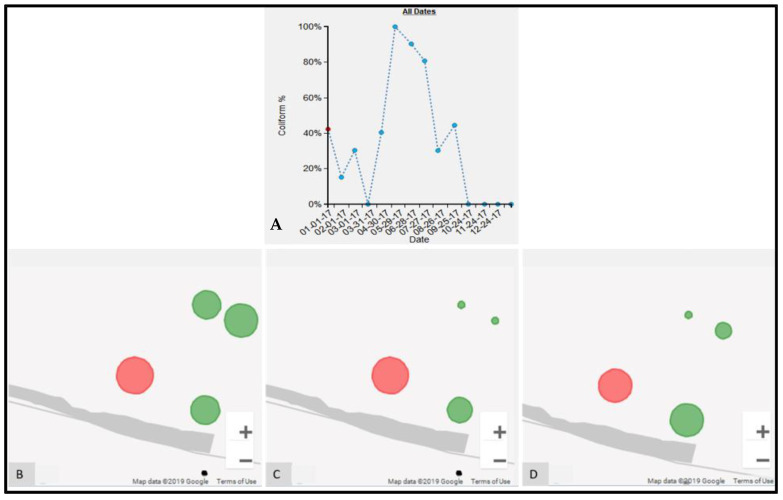
FC count data for water point S17 in S neighborhood. (**A**) Time-chart for S17 water point, the red ellipse indicates periods of high FC count; (**B**) proportional symbol map for FC count for May; (**C**) proportional symbol map for FC count for June; (**D**) proportional symbol map for FC count for July.

**Figure 15 ijerph-19-08902-f015:**
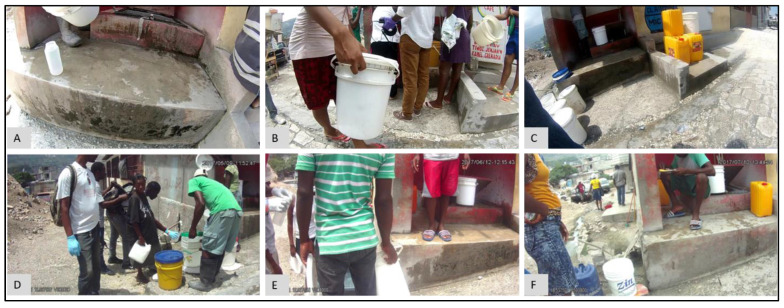
SV images for water point S17 for: (**A**) February, (**B**) August, (**C**) September, (**D**) May, (**E**) June, and (**F**) July 2017.

**Figure 16 ijerph-19-08902-f016:**
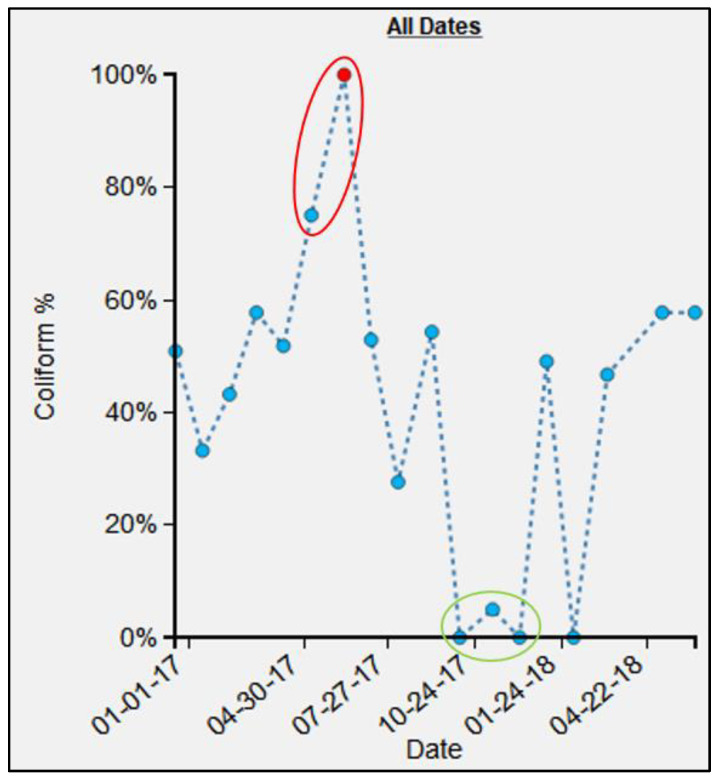
Epidemiological time-chart indicating FC count for P1. The red ellipse indicates consistently higher FC count values, whereas the green ellipse indicates consistently lower FC count value.

**Figure 17 ijerph-19-08902-f017:**
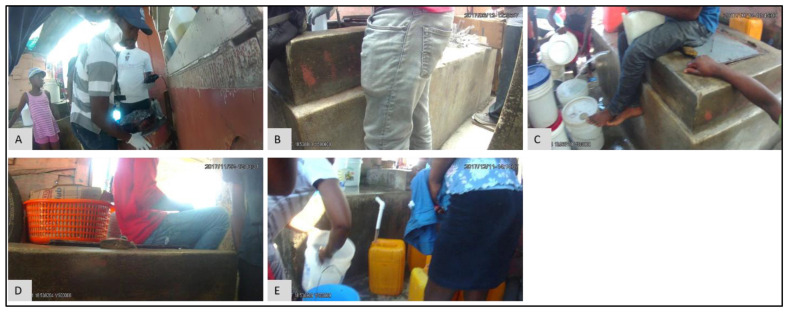
SV images for water point P1 for: (**A**) May, (**B**) June, (**C**) October, (**D**) November, and (**E**) December 2017.

## Data Availability

The datasets that have been used for this study are deemed confidential and cannot be shared.

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
