# Peer review of "Spatial Video and EpiExplorer: A Field Strategy to Contextualize Enteric Disease Risk in Slum Environments"

_ijerph, 2022, doi:10.3390/ijerph19158902_

Round 1
Reviewer 1 Report
Thank you for your innovative work with this unique data set and country context. Overall very well written. I have offered a number of detailed suggestions to further strengthen your article, for your review and consideration. I look forward to reading your revision.
1. Abstract
The abstract section is not clear what the main purpose is? How did you collect the data?
1. Introduction
The objectives are not clearly identified what gap did you provide to investigate (see line 48-89).
2. Design
Design section not clear what and how did you apply. Separate data setting is required.
Line 146-158 is not clear
Line 224-232 is not clear
3. Findings
3. An empirical.... changed into the findings or (3. Findings) and then added sub-section
Line 328-344 (included figure 15-16) should be added more information.
4. Discussion
Discussion section is too limited debate/discussion with the main findings.
Line 346-407 should be discussed with main findings and then discussed with the other scholars are required. This is because I found only explained the findings, no discussion with main findings or others.
* Should be added more information about (1. Practical implications, 2. Limitation and further study)
5. Conclusion
Should be revised all sentience because is not related to the main findings (see line 409-416).
* Many grammar errors, provided native edits/proofreads are before re-submission is required.
Author Response
1. Abstract
The abstract section is not clear what the main purpose is? How did you collect the data?
The abstract has now been considerably rewritten to more clearly state the purpose of the paper including a brief introduction of the data being collected / analyzed.
More specifically:
In this paper we use bespoke spatial programming to create a framework for flexible fine scale exploratory investigations of simultaneously collected water quality and environmental surveys in three different informal settlements of Port-au-Prince, Haiti. We dynamically mine these spatio-temporal epidemiological and environmental data to provide insights not easily achievable using more traditional spatial software such as geographic information system (GIS). Results include sub neighborhood maps of localized risk that vary month. Most interestingly some of these epidemiological variations might have previously been erroneously explained because of proximate environmental factors and/or meteorological conditions.
1. Introduction
The objectives are not clearly identified what gap did you provide to investigate (see line 48-89).
The introduction paragraphs have now been cleaned to more directly show the objectives of the paper. More specifically:
In this paper we address this localized mapping conundrum by drawing on the types of data analytics required to support a more spatially exploratory approach. During the authors spatial response to Covid-19, the power of real time data analytics and “big data” investigations, especially the use of dashboard style data manipulations, led to a fresh perspective on epidemiological “space-time risk scenarios” [18]. While other interactive software are available to interactively explore data [19], including different dashboard applications which allow for the inclusion of spatial data [20], none have the location specific sophistication and flexibility needed to fully leverage the different data types generated here. In this paper we show how exploratory spatial data analytics utilizing bespoke spatial programming can be used to more effectively answer (epidemiological) questions based on the types of data available, interactive multi-media visualizations, and the environment being studied. To illustrate the power of this approach we explore the complex interactions in the epidemiological, environmental and human behavior risks around water access and drainage locations for three coastal informal settlements in Port-au-Prince, Haiti.
2. Design
Design section not clear what and how did you apply. Separate data setting is required.
Revamped the design section. Added a separate data section for the Case Study.
Line 146-158 is not clear
Changed for more clarity
Line 224-232 is not clear
Changed for more clarity
3. Findings
3. An empirical.... changed into the findings or (3. Findings) and then added sub-section
Changed this section as case study and included the results. The discussion about the results have been added to the discussion section
Line 328-344 (included figure 15-16) should be added more information.
Moved to discussion section and added more details
4. Discussion
Discussion section is too limited debate/discussion with the main findings.
Discussion section is revamped to include more insights
Line 346-407 should be discussed with main findings and then discussed with the other scholars are required. This is because I found only explained the findings, no discussion with main findings or others.
More details added in discussion section.
* Should be added more information about (1. Practical implications, 2. Limitation and further study)
Clearly defined the limitations as well as future works in separate paragraphs
- Conclusion
Should be revised all sentience because is not related to the main findings (see line 409-416)
Revamped the conclusion section.
* Many grammar errors, provided native edits/proofreads are before re-submission is required.
Checked for grammar errors and proofread to remove minor errors.
Reviewer 2 Report
Interactive features for the exploration of water quality offer high added value, even though the paper offers little methodological progress. In high-income countries, for example, the inspection of wastewater sewer systems has been done for decades with georeferenced videos, that have been fed into analysis systems. Nevertheless, having the proposed system available would be of great practical importance for water quality analysis in low-income countries. In particular, its ease of use provides great value, even for non-specialist users.
The title may lead the reader in another direction. In any case, I had expected three-dimensionality as Spatial Video (see https://spatial.io/ for instance)
Lines 135-144. For illustration purposes, would it be possible to provide sample records for the database tables?
Figure 2 What do the green, yellow, and blue marker pins mean? Explain.
Figure 3, 4, and following. It remains unclear what exactly the horizontal axis of the graphs mean. For example, in Figure 3, the text seems to indicate that the horizontal axis represents individual instances. However, the graph annotation reads "date" without indicating a time scale. All corresponding figures should be revised to include a clear (time or other) scale and annotation for the horizontal axis. A very good example of how to do it is Figure A1B
Author Response
Lines 135-144. For illustration purposes, would it be possible to provide sample records for the database tables?
Thank you for the suggestion. This is added as a figure (Figure 2).
Figure 2 What do the green, yellow, and blue marker pins mean? Explain.
The figure caption now has the details.
Figure 3, 4, and following. It remains unclear what exactly the horizontal axis of the graphs mean. For example, in Figure 3, the text seems to indicate that the horizontal axis represents individual instances. However, the graph annotation reads "date" without indicating a time scale. All corresponding figures should be revised to include a clear (time or other) scale and annotation for the horizontal axis. A very good example of how to do it is Figure A1B
Thank you for the suggestion. Changed the chart generation logic in the software to accommodate axis ticks.
Reviewer 3 Report
It is an interesting paper that could be helpful to the inhabitants of Haiti and others countries that have the same problems. A simple idea that brings together GIS and measurements can help many people in a water shortage situation
Maybe a little introduction to the recent history of Haiti can be enlightening about the water problem that occurs in Haiti.
Author Response
Maybe a little introduction to the recent history of Haiti can be enlightening about the water problem that occurs in Haiti.
Added this to the Case Study Section
Round 2
Reviewer 1 Report
Read well than previous version and all comments are revised.